# DNA Replication Through Strand Displacement During Lagging Strand DNA Synthesis in *Saccharomyces cerevisiae*

**DOI:** 10.3390/genes10020167

**Published:** 2019-02-21

**Authors:** Michele Giannattasio, Dana Branzei

**Affiliations:** 1IFOM (Fondazione Istituto FIRC di Oncologia Molecolare), 20139 Milan, Italy; 2Dipartimento di Oncologia ed Emato-Oncologia, Universita’ degli Studi di Milano, 20122 Milan, Italy; 3Istituto di Genetica Molecolare, Consiglio Nazionale delle Ricerche (IGM-CNR), Via Abbiategrasso 207, 27100 Pavia, Italy

**Keywords:** DNA replication, lagging strand DNA synthesis, Okazaki fragment processing, strand displacement DNA synthesis, DNA helicases, flap endonucleases

## Abstract

This review discusses a set of experimental results that support the existence of extended strand displacement events during budding yeast lagging strand DNA synthesis. Starting from introducing the mechanisms and factors involved in leading and lagging strand DNA synthesis and some aspects of the architecture of the eukaryotic replisome, we discuss studies on bacterial, bacteriophage and viral DNA polymerases with potent strand displacement activities. We describe proposed pathways of Okazaki fragment processing via short and long flaps, with a focus on experimental results obtained in *Saccharomyces cerevisiae* that suggest the existence of frequent and extended strand displacement events during eukaryotic lagging strand DNA synthesis, and comment on their implications for genome integrity.

## 1. The Replication Fork and the DNA Synthesis Apparatus

With the scope of introducing the proteins involved in lagging strand DNA synthesis in the context of the replication fork and replisome, we begin with a brief introduction of DNA replication initiation. However, due to the limited extent of this introduction, we re-direct readers interested in the mechanisms and regulations of DNA replication origin activation, replisome assembly and structure to recent reviews on this topic [1,2,3,4]. DNA replication initiates from specific regions on the chromosomes, known as origins of replication, which are well defined in *Saccharomyces cerevisiae* by the presence of an autonomously replicating sequence (ARS), but are less defined in vertebrates [3,5,6,7,8]. Briefly, origin licensing in *S. cerevisiae* starts in M/G1 with the loading of the MCM2-7 (Minichromosome Maintenance Complex) complex on an ARS sequence bound by ORC (Origin Recognition Complex). For DNA replication to initiate, an inactive double hexamer of Mcm2-7 needs to be activated and separated into single hexamers, each of them in complex with Cdc45 (Cell division cycle 45) and GINS (GO, Ichi, Ni and San complex), to form the replicative CMG (Cdc45-Mcm2-7-GINS) helicase [1]. Formation of an active CMG complex is promoted by S-CDK (Cyclin Dependent Kinase)-dependent phosphorylation of Sld2 (Synthetic Lethal with Dpb11-1 number 2) and Sld3 (Synthetic Lethal with Dpb11-1 number 3) [9], and by DDK (Dbf4-dependent kinase)-dependent phosphorylation of MCM [3,10,11]. MCM appears to be the only essential DDK target for replisome assembly and origin activation in an in vitro system reproducing DNA replication origin firing using purified *S. cerevisiae* proteins [12]. DDK phosphorylation of Mcm4/6 generates binding sites for Sld3 [13], which recruits Cdc45. Phosphorylated MCM acts together with CDK-phosphorylated Sld2 and Sld3 to trigger the recruitment of firing factors, which remodel the MCM double hexamer to form the CMG helicase [5]. How this remodeling occurs is not fully understood [14]. 

DNA polymerase ε is recruited to MCM through Dpb11 (DNA polymerase B possible subunit 11)-mediated interactions with CDK-phosphorylated Sld3 and Sld2, while Cdc45 recruitment to MCM requires DDK-dependent phosphorylation of MCM [3,5]. Ctf4 (Chromosome Transmission Fidelity 4) and MCM10 participate in the recruitment of DNA polymerase α to chromatin and to the CMG complex for initiation of DNA replication [15,16]. MCM10 also has a crucial role in the activation of the CMG helicase for replication initiation [12,14]. DNA polymerase α also interacts directly with MCM [17], but the detailed mechanism of Pol α recruitment to the replisome is not fully elucidated. Furthermore, evidence from both in vitro [12,18] and in vivo [19] systems indicate that Ctf4 is dispensable for Pol α-Primase -mediated priming. Rather, it seems that Ctf4 is important for parental histones transfer to the newly replicated lagging strand filament [20]. DNA polymerase δ is recruited to the replisome through interaction with PCNA (proliferating cell nuclear antigen), which is loaded on the growing 3’ ends of the newly synthesized filaments by the replication factor C (RFC) [7,21]. In Figure 1, a schematized view of a replication fork and the replisome is shown. The antiparallel nature of the DNA double helix and 5’–3’ direction of the DNA synthesis at the replication fork impose that one newly synthesized strand of the replication fork is polymerized in a continuous way (leading strand filament), while the other is subjected to discontinuous DNA synthesis (lagging strand filament) [22]. Upon DDK and CDK-mediated phosphorylation events, MCM double hexamers get activated, the DNA at the origin is unwound and Okazaki fragments composed of RNA and DNA initiator primers (DNAi) are synthesized both on the leading and lagging strands [7,23] by DNA polymerase α-Primase (Figure 1). 

It was recently demonstrated that during the first steps of the leading strand establishment, DNA polymerase δ substitutes Pol α and contributes to the initial elongation of the nascent leading strand before being replaced by Pol ε [24]. Multiple in vitro systems support the idea that Pol δ synthesizes the lagging strand while Pol ε the leading strand [22,24,25,26,27,28]. However, genetic evidence suggests that Pol δ could perform replication of a large part of chromosomal DNA acting either as lagging or leading strand DNA polymerase [8,29]. 

While the mechanisms of Ctf4 and Pol α loading at the replication forks are not fully understood, the switch between Pol α and DNA polymerase δ (or DNA polymerase ε) is thought to be triggered by the intrinsic lack of processivity of Pol α. Pol α falls off from the growing 3’ ends after having synthesized the RNA-DNA initiators. At that time, the newly synthesized strands are bound by the RFC (Replication factor C) complex, which will load PCNA (Proliferating Cell Nuclear Antigen) and the two processive DNA polymerases δ and ε on the lagging and leading strands, respectively [6,30,31] (Figure 1). 

The pausing complex composed by Mrc1-Csm3-Tof1 (Mediator of the Replication Checkpoint 1, Chromosome Segregation in Meiosis 3, Topoisomerase Interacting Factor 1) [32,33] represents a structural component of the replisome (Figure 1). This complex is important to facilitate normal rates of fork progression and, upon replication fork slow-down, it collaborates with DNA polymerase ε to activate the S-phase checkpoint [33,34]. The Mrc1-Csm3-Tof1 complex also facilitates fork rotation to convert positive super-coiling ahead of the fork to DNA catenanes behind the fork, thus facilitating fork progression [35]. In order for the elongation phase of DNA synthesis to proceed in a processive way, DNA topoisomerases are necessary to relieve DNA positive super-coiling that accumulates ahead of the moving replication fork and DNA catenation induced behind the fork [36]. 

The lagging strand filament is composed of multiple Okazaki fragments (OFs) each containing a piece of RNA initiator (RNAit) at its 5’ end. The RNAit has an average length of 20 nt in *E. coli* [37,38] and 10 nt in *S. cerevisiae* (Figure 1) [39,40]. The OFs undergo a complex process called OF maturation during which the RNAits are removed, and the resulting DNA nicks are sealed creating a continuous lagging strand filament and a fully replicated DNA molecule [41,42]. If the process of OF maturation is not complete and DNA flaps or nicks are left in the genome, deletions, amplifications of DNA sequences and double strand breaks (DSBs) can arise [43]. OF maturation involves many factors and enzymatic activities. Specifically, PCNA, the proof reading activity and strand displacement activity of DNA polymerase δ, human FEN1 (Flap structure specific ENdonuclease I) corresponding to budding yeast Rad27 (Radiation sensitive 27), Dna2 (DNA replication Helicase/Nuclease 2), Pif1 (Petit integration frequency 1) helicase, RNase H2, Exo1 (Exonuclease I) and DNA ligase 1 participate in the processing of OFs [39,40]. Table 1 summarizes the protein factors and the corresponding genes involved in OF maturation with their nomenclature in *S. cerevisiae, Schizosaccharomyces pombe* and mammals.

A schematic representation of the pathways and reactions involved in OF processing is shown in Figure 2. The key step in OF maturation is the removal of the RNAit primer (and likely also of the DNA initiator fragment synthesized by Pol α) and the creation of a ligatable nick, which is a ssDNA nick with the 3’ and 5’ ends close to each other so that they can become a substrate for the DNA ligase I [44]. The ligatable nicks during OF processing could be created by: (i) direct removal of the RNA primer by RNase H2 [45,46] and limited Pol δ- mediated DNA synthesis (Figure 2A); (ii) by idling of DNA polymerase δ, a process in which its proof-reading activity is balanced by limited strand displacement activity, FEN1-mediated cutting of the resulting short DNA flaps and DNA ligation [47,48] (Figure 2B,C); (iii) by limited Pol δ-mediated strand displacement and Fen1- or Exo1-dependent processing of the resulting DNA flap (Nick translation reaction) [49] (Figure 2B,C); (iv) by extended Pif1-PCNA-Pol δ-mediated strand displacement and subsequent cut of the flap by Dna2 and FEN1 [50,51,52,53] (Figure 2D), or by DNA recombination mechanisms in the absence of Fen1 [54] (Figure 2E) (see below). The regulation, frequency and crosstalk between these different OF processing pathways are largely unknown. According to the above models, absence of Dna2 would induce the formation of long DNA flaps that cannot be processed by Fen1. DNA flaps with long ssDNA tails will be bound by the replication protein A (RPA) and ssDNA-RPA complexes will act as platforms to recruit and activate the apical kinase scMec1 (human ATR, spRad3), triggering the DNA Damage Response (DDR) [55,56]. Importantly, the extent of DDR activation correlates with the amount and length of ssDNA-RPA complexes present in the cells [56]. Persistent DDR hyper-activation due to a cellular over-load of DNA flaps with long ssDNA tails is expected to induce cell cycle arrest and cell death. Accordingly, the lethality of Dna2 depleted cells was shown to be due to DDR activation (see below) [57,58,59]. On the contrary, in the absence of Fen1, long flaps created by extended Pif1-PCNA-Pol δ-mediated strand displacement would be shortened by Dna2, so that short flaps can become substrates for Exo1 (or for homologous recombination). According to these models, mild DDR activation is detectable in *rad27* cells (see below and Figure 2F,G) and these cells are viable. For the same reasons, *exo1*Δ cells are alive because Fen1/Rad27, and possibly RNase H2, process the flaps that are shortened by Dna2. Ligation of the nicks by DNA ligase I at the end of OF maturation is an essential step. *S. cerevisiae* cells deleted for *CDC9* (Cell Division Cycle 9) that encodes DNA ligase I, do not support viability [44]. Rad27/FEN1 [60,61] belongs to the Rad2 (Radiation sensitive 2)/XPG (Xeroderma Pigmentosum complementation group C) family of nucleases, characterized by having 5’ flap endo-nuclease and 5’–3’ exonuclease activity [62,63]. FEN1 interacts directly with and is stimulated by PCNA [64,65,66,67,68], and is necessary for OF processing during DNA replication [61,69,70,71]. Dna2 is an essential 5’–3’ DNA helicase, 5’–3’ directed ssDNA translocase, 5’ flap endonuclease and 5’–3’ exonuclease [72,73,74,75] that interacts physically and genetically with FEN1/Rad27 [76,77], PCNA [78] and DNA polymerase δ [77]. Because of its biochemical activities and interactions with FEN1 and PCNA, it was proposed that Dna2 participates in the processing of OFs. Exo1 is a Rad2/XPG endo-exonuclease involved in mismatch repair, double strand break repair and OF processing [79]. Besides DNA polymerase δ, PCNA and DNA ligase I, among all the other factors that participate in OF processing in *S. cerevisiae*, Dna2 is the only one that is essential [73,80], although the reasons of its essentiality are still under debate. 

Budding yeast RNase H2 is a non-essential heterotrimeric complex [81]. The catalytic subunit RNase H2 subunit A (also called RNase H35, encoded by the gene *RNH35/RNH201* [82]), is related to the prokaryotic RNase HII [83] and mammalian RNase HI [84]. RnaseH2 interacts with PCNA [85,86] and has an RNA-DNA endo-ribonuclease activity that hydrolizes the phosphate bond at the 5’ side of a ribonucleotide-deoxyribonucleotide sequence within RNA-DNA duplexes [81]. The other two non-catalytic subunits of RNase H2 (subunit B and C) are encoded, respectively, by the genes *RNH202* and *RNH203* [81]. Notably, the two small non-catalytic subunits of RNAse H2 are necessary for the functionality of the enzyme [81]. Because of its activity on DNA duplexes carrying ribonucleotides, RNase H2 has been proposed to participate in the removal of the RNAits from OFs (Figure 2A) [42]. While *rnh35* and *rad27* cells are alive, although slowly growing, the double mutant *rnh35 rad27* is dead [46,60,87,88]. Exo1 is a non-essential nuclease that interacts with and is stimulated by PCNA [79,89]. *RAD27* deletion induces cell lethality when combined with point mutations that specifically inactivate the 5’-flap endonuclease activity of Exo1, but not with alleles that inactivate its 5’-3’ directed exonuclease activity suggesting that DNA flaps in *rad27* cells are cut by Exo1 [90]. Since *exo1 rnh35* cells are alive [91], one possible explanation is that Rad27 and Exo1-RNaseH2 constitute two redundant pathways of OF maturation. Considering these genetic interactions and the model presented in Figure 2, it still remains to be explained why long flaps can be shortened in the absence of Rad27, Exo1 or RNase H2, possibly by Dna2, to a level compatible with cell viability while the ones in the *rad27 exo1* or *rad27 rnh35* induce cell lethality even if Dna2 (and homologous recombination) are envisaged to be functional in these cells. One possibility is that either the Rad27 or Exo1-RNAse H2 pathways act redundantly to modify the DNA flaps so that they can become substrates for Dna2 (or homologous recombination-see below). In this scenario, when these two pathways are inactivated (as in *rad27 rnh35* or *rad27 exo1* mutant cells), DNA flaps become refractory to processing, accumulate in the cells and lead to DDR-dependent cell death. According to this view, it would be interesting to analyze, in single cell cycle experiments, the DNA structures accumulating in cells deprived of Rad27 and Exo1 or Rad27 and RNase H2. A further complexity in the mechanism of OF maturation comes from the fact that *RAD27* deletion shows synthetic lethality when combined with mutations in genes involved in homologous recombination (HR) [54], supporting the idea that several pathways back up Fen1 during OF maturation (Figure 2) and that DNA structures generated during defective OF maturation are processed by HR. In the same way, cells defective for RNase H2 rely on homologous recombination to support viability, although it is not clear if this negative genetic interaction reflects a more general role of RNase H2 in counteracting the accumulation of ribonucleotides inserted in the genome outside the context of OFs [92]. Taken together, these genetic results suggest that FEN1, RNase H2 and Exo1 have overlapping roles in OF processing. Importantly, although *DNA2* is an essential gene, the use of synthetic genetic array (SGA) screens with point mutants of *DNA2* revealed that the *dna2-1* mutant (carrying a P504S substitution that significantly decreases the ATPase, DNA helicase and nuclease activities of the protein) is synthetic lethal with *RAD52* (Radiation sensitive 52), *RNH35* and *RAD27* deletions, while the *dna2-2* mutant (carrying an R1235Q substitution in the helicase domain IV that has a less severe impact on the nuclease activity of the protein) is synthetic lethal with *EXO1* deletion and with the *pol3-01* allele, a Pol δ proof reading mutant with increased strand displacement potential [93]. These genetic interactions suggest that increased strand displacement activity of Pol δ is deleterious when Dna2 is not functional and that defective Dna2 functions are backed up by Rad27, HR and RNase H2. Indeed, the negative genetic interactions of *dna2* alleles are identical to the ones identified for the *RAD27* deletion, suggesting that Dna2 nuclease and Rad27/Fen1 have compensatory roles in OF maturation. More generally, the genetic interactions between mutations in the genes involved in OF maturation highlight the notion that this process is carried out by an intricate network of pathways (Figure 2). 

Pif1 is a ssDNA-dependent ATPase, 5’–3’ directed DNA helicase and ssDNA translocase that interacts with PCNA and is required to assist DNA replication across natural pausing sites and specific DNA sequences, such as G4 DNA [94,95,96,97,98,99]. Pif1 interacts with Fen1 [100] and its inactivation in budding and fission yeast suppresses, respectively, the cell lethality associated with loss of Dna2 or temperature sensitive alleles of *cdc24* [53,59]. Cdc24 (cell division cycle 24) is an essential protein that participates with DNA polymerase δ to the lagging strand DNA synthesis in *S. pombe* [101]. Cdc24 interacts physically with Dna2 and also has roles in the Dna2-dependent long-range resection branch during DSB repair through HR [101,102]. Cdc24 depletion is thought to inactivate Dna2 functions in lagging strand DNA replication and DSB repair. Based on these findings, it was proposed that during OF processing, Pif1 creates a sub-population of long DNA flaps that must be processed by Dna2 (see below) [53], although the 5’–3’ directed DNA helicase activity of Pif1 would peel off the 3’ end of the newly synthesized fragment instead of displacing the 5’ end of the last OF [103,104]. This apparent discrepancy between the 5’–3’ polarity of Pif1 helicase and the processing of the 5’ ends of OFs can be rationalized if Pif1 is considered part of a strand displacement “machine” composed by PCNA-Pol δ-Pif1 (see below). In this case, Pif1 acts as a strand displacement co-factor of the PCNA-Pol δ complex and it utilizes its ATPase and 5’–3’ ssDNA translocase activity to translocate on the displaced 5’ end of the OF (see below). Subsequent work showed that the lethality of *dna2* mutants is due to the activation of the DNA damage response (DDR) by RPA-coated, long single stranded DNA flaps [57]. According to this interpretation, recent work using transmission electron microscopy to visualize in vivo DNA replication intermediates showed accumulation of long DNA flaps in *S. pombe* spores deleted for *dna2* or after conditional depletion of Dna2 in *S. cerevisiae* [58,105]. Although the frequency of the long DNA flaps is similar in the two reports, there is still an open debate about the average length of the ssDNA tails in the DNA flaps in Dna2 depleted cells in unperturbed conditions. Interestingly, both reports find a consistent fraction of DNA flaps at a big distance from the DNA replication fork branching point. In electron microscopy, the DNA replication fork branching point is defined as the point in which the parental strand and the two newly replicated strands get in contact creating the Y-like fork structure. This result suggests that DNA flaps could be processed outside the context of the moving leading and lagging strand apparatus at the DNA replication fork branching point [58,105]. This interpretation is in agreement with a recent report that shows that OF maturation can be uncoupled from ongoing DNA synthesis at replication forks [106]. This in vivo evidence supports the idea that extended strand displacement events occur during lagging strand DNA synthesis and, as a consequence, DNA flaps with long ssDNA tails can be created in the cells in the absence of the factors responsible for their processing. A logical implication of these findings is that DNA flaps formed in the absence of Rad27, Exo1 or RNase H2 do not induce checkpoint hyper-activation to a level that is incompatible with cellular proliferation, whereas the long DNA flaps produced in the absence of Dna2, although less frequent, cause hyper-activation of DDR [57,59]. The most reasonable explanation is that the large majority of the short flaps are processed via limited DNA polymerase δ strand displacement activity, followed by FEN1, Exo1-RNase H2 mediated processing, while less frequent long DNA flaps require the activity of Dna2 to be processed (Figure 2). The genetic interactions reported above in this manuscript are also compatible with a linear pathway in which all the DNA flaps must undergo FEN1 or Exo1-RNase H2-mediated processing before being further processed by Dna2 or HR (see next paragraphs). Dynamic interaction of DNA polymerase δ proof reading and strand displacement activities are regulated by PCNA and FEN1 interactions and are important in the processing of the OFs [40,107]. Accordingly, mutations in Pol δ 3’–5’ proofreading activity causing increased strand displacement are lethal in combination with *RAD27* deletion [108], suggesting that the proof-reading activity of Pol δ can compensate for Fen1 during OF maturation. Recent in vitro studies using purified Pol δ showed that the role of the 3’–5’ proof reading activity of Pol δ in OF maturation is to counteract extended Pol δ-dependent strand displacement of the 5’end of the last OF in a process called “DNA polymerase δ idling”, which prevents the creation of long flaps and favors thereby FEN1- or Exo1-mediated processing, thus contributing to direct maturation of OFs toward DNA ligation [40,47] (Figure 2B,C). On the other hand, the processivity of DNA polymerase δ and its strand displacement activity are stimulated by interaction with PCNA and by mutations that abolish its proof-reading activity [48,109,110]. While Pol δ is not efficient in making strand displacement reactions in vitro, mutations in its 3’–5’ exonuclease activity stimulate its strand displacement potential, which is inhibited by long ssDNA tails in the resulting DNA flaps [40,48]. Importantly, if PCNA is present in the strand displacement in vitro reactions, Pol δ carries out efficient strand displacement, which becomes insensitive to the inhibitory effect of the growing ssDNA tails in the DNA flaps [48]. This biochemical evidence suggests that the presence of PCNA enhances the strand displacement potential of Pol δ either by masking inhibitory sites on the enzyme that can bind ssDNA or by enhancing Pol δ processivity. When Pif1 is added to in vitro reactions simulating OF processing, the length of Pif1-PCNA-Pol δ−displaced flaps increases to 30-110 nt [50,51,52] supporting the idea that Pif1 could work as a sort of strand displacement accessory factor for Pol δ. In this direction, in other in vitro systems more suitable to detect extended strand displacement events, the complex Pif1-PCNA-Pol δ can support strand displacement DNA synthesis for 2.9 kilobase pairs (kbps) on a duplex linear DNA carrying a DNA flap or for 10 kbps on a circular duplex plasmid carrying a ssDNA gap [94,98].

## 2. Strand Displacement Activity of *Escherichia coli*, Bacteriophage and Viral DNA Polymerases

A key step in the maturation of OFs during lagging strand DNA synthesis in *S. cerevisiae* is the Pol δ-mediated strand displacement (Figure 2). However, the in vivo frequency, extent and regulation of this event are not known. Historically, the strand displacement activity of DNA polymerases was discovered during the study of the biochemical properties of bacterial, bacteriophage and viral DNA polymerases. We review a set of results that highlight the existence of potent strand displacement activities of prokaryotic, bacteriophage and viral DNA polymerases. Early in vitro studies showed that intact dsDNA molecules and closed circular dsDNA cannot be used as templates for DNA synthesis by the *E. coli* DNA polymerase I because of the lack of extensible 3’-hydroxyl ends juxtaposed to a template DNA strand [111,112]. On the contrary, these molecules can serve as templates for DNA polymerization when they contain ssDNA breaks (nicks) with extensible 3’-hydroxyl termini [111,112]. These early biochemical reactions showed that when a dsDNA molecule carrying a nick is used as template by *E. coli* DNA polymerase I there is a “first phase” of DNA synthesis in which the 3’-hydroxyl terminus in the nick is extended through 5’–3’ directed DNA polymerization, which is accompanied by a robust DNA polymerase I-dependent 5’–3’ directed exonucleolytic degradation of the 5’ end of the nick [111,112]. In this first phase of DNA synthesis through a nick, the extent of the DNA polymerization equals the extent of the hydrolysis of the 5’ end of the nick. Thus, there is no net DNA synthesis and the position of the nick is translated in a process called “nick translation” [111,112] (Figure 3A). The length of the DNA polymerized/hydrolyzed in this “first nick translation phase” of DNA synthesis is limited [111,112]. The existence of a strand displacement activity of the *E. coli* DNA polymerase I was hypothesized when it was noted that this “first nick translation phase” is followed by a “second phase” of DNA polymerization in which the 5’–3’ exonucleolytic digestion of the 5’ end of the nick is terminated, the 5’ end of the nick starts to be preserved and a net DNA polymerization is detectable [111,112]. Importantly, this second phase of DNA synthesis through strand displacement is not prevented by the inactivation of the 5’–3’ exonuclease activity of DNA polymerase I supporting the idea that, at a certain point, DNA polymerase I starts to displace the 5’ end of the nick, which is annealed to the template strand [112] (Figure 3A). This activity was defined as DNA synthesis through strand displacement. It is not known if there are underlying structural determinants for the switch between the nick translation phase and the strand displacement phase or if this switch is regulated in vivo. 

Other early studies showed that the DNA polymerase of the T4 bacteriophage (T4 DNA polymerase) together with the gp32 ssDNA binding protein and helix-destabilizing factor can sustain DNA synthesis through strand displacement on a nicked duplex T7 DNA template leading to the formation of branched DNA molecules seen through transmission electron microscopy (TEM) [113]. Further electron microscopy analysis confirmed that the proteins encoded by the T4 genes 43 (gp43-T4 DNA polymerase), 32 (gp32 ssDNA binding protein), 44 and 62 (gp44/62 structural complex), 45 (gp45 DNA polymerase accessory factor), can carry out extended DNA synthesis through strand displacement on nicked duplex circular ColE1 plasmid DNA or nicked linear duplex T4 genomic DNA and that the major products of the strand displacement synthesis are double strand (ds) circular or linear DNA molecules with very long displaced single strand DNA tails [114] (Figure 3B). Interestingly, this “five proteins system” does not contain the gp41 hexameric DNA helicase of T4 suggesting that a DNA polymerase, the ssDNA binding protein and some structural accessory factors are sufficient to carry out extended and efficient strand displacement DNA synthesis [114]. Subsequent experiments showed that also the DNA polymerase of the T7 bacteriophage promotes DNA synthesis through strand displacement on a duplex circular pBR322 plasmid molecule containing one ssDNA nick [115]. Interestingly, in the same study, electron microscopy analysis of the products of the T7 DNA polymerase-dependent strand displacement DNA synthesis revealed the presence of duplex DNA molecules containing dsDNA branches [115]. It was proposed that the DNA flaps generated during DNA synthesis through strand displacement could undergo several types of isomerization reactions (also involving re-initiation of the DNA synthesis on the isomerized templates) leading to the formation of branched dsDNA molecules [115]. Although these hypothetical isomerizations do occur in vitro, it is not known if they happen in vivo. Importantly, frequent products of in vivo and in vitro DNA synthesis through strand displacement are flap molecules with long displaced ssDNA tails. Of interest, a more recent study showed that a “two-protein” system composed of the T4 DNA polymerase and the gp41 DNA helicase is sufficient to sustain a processive, extended, fast (400–500 bp/s) and ATP-dependent strand displacement activity [116]. The genome of the bacteriophage 𝜙29 is a 19 kb linear dsDNA molecule with one TP protein molecule covalently bound at each of its end [117]. TP-DNA is replicated through strand displacement DNA synthesis by the 𝜙29 DNA polymerase, which uses the TP protein as a primer attaching the first nucleotide to one of its hydroxyl group in a process defined as protein-primed DNA synthesis through strand displacement [118]. Electron microscopy analysis of the products of in vitro DNA synthesis reactions or DNA replication intermediates isolated form in vivo 𝜙29 DNA replication allowed to visualize the existence of DNA flaps with long displaced ssDNA tails [119,120,121]. The p6 ssDNA binding protein of 𝜙29 was shown to have a key role in the DNA synthesis through strand displacement either because it stabilizes the displaced ssDNA tails during DNA synthesis through strand displacement or because it has a double helix de-stabilizing activity, which favors the advancement of the DNA polymerase on the template and the displacement of the 5’ end encountered by the DNA polymerase during DNA synthesis through strand displacement [119].

These results led to the concept that protein modules composed of a potent DNA polymerase activity coupled to a DNA helicase (or to a ssDNA binding protein) can promote processive, fast and extended strand displacement DNA synthesis leading to the formation of DNA molecules containing DNA flaps with long protruding ssDNA tails. Although bacteriophage DNA polymerases T4, T7 and 𝜙29 show potent strand displacement activity when combined with the corresponding ssDNA binding proteins (or DNA helicases), there are also examples in nature of viral DNA polymerases that can carry out potent strand displacement activities without accessory factors. For example, the Epstein-Barr virus DNA polymerase holoenzyme (BALF5-BMRF1) is able by itself to carry out strand displacement DNA synthesis in vitro to produce circular duplex DNA molecules with long protruding ssDNA tails [122]. Like DNA polymerase δ, also other eukaryotic DNA polymerases can carry out extended DNA synthesis through strand displacement. For example, the calf thymus DNA polymerase β can use duplex circular DNA carrying a ssDNA gap as template and synthesizes large sequences of DNA by displacing the 5’ end of the gap through strand displacement DNA synthesis [123].

Other in vitro reactions show that DNA polymerase β-mediated strand displacement, which is thought to be an essential step of the long patch base excision repair pathway in eukaryotes is stimulated by PARP-1 (Poly(ADP-Ribose) Polymerase 1) [124]. PARP1 catalyzes the transfer of poly(ADP-Ribose) chains to proteins, DNA and small chemical groups [125]. Although PARP1 has a role in fork restart, DNA repair of ssDNA breaks and in transcription regulation, its exact biochemical function in these processes remains elusive. Importantly, it has been shown that mammalian cancer cells defective in HR-mediated DNA repair of DSBs and/or fork stabilization rely on PARP1 for survival, while non-defective cells do not. This led to the conceptual framework of usage of PARP1 inhibitors as chemotherapeutic agents to eradicate selectively breast and ovarian cancer cells defective in HR [126]. Intriguingly, it has been recently shown that PARP regulates DNA replication fork speed but the molecular mechanisms of this regulation are still elusive [127]. The strand displacement activity of a DNA polymerase appears to be inversely correlated to its propensity to slippage during DNA replication [128]. Since DNA polymerase slippage is a replication error, which can lead to deletions or amplifications of DNA sequences [129,130], it is possible that DNA polymerases with strong strand displacement activity synthesize DNA with higher fidelity. 

## 3. Limited Pol δ Strand Displacement Coupled with Short Flap Pathway of Okazaki Fragment Processing in *S. cerevisiae*

The extent of Pol δ-dependent strand displacement is a key step at the crossroads between short and long flap processing of OFs. If strand displacement is limited, OF maturation is directed towards the short flap pathway, while a more extended strand displacement would create longer flaps that need to be processed by Dna2 [76,131,132] (see Figure 2). Importantly, the definition of long and short flap pathways is only theoretical. In wild type conditions, ssDNA tails generated during OF processing are immediately processed and may never be exposed, thus avoiding RPA binding to ssDNA and futile DDR activation coupled with cell cycle arrest [56]. The current models of OF maturation predict that the flaps are processed, mostly by Fen1 (limited strand displacement) or by a sequential action of Dna2 and Fen1 (more extended strand displacement) immediately after Pol δ has created them [39,40,107] (Figure 2). OF maturation (in wild type conditions) can therefore be considered similar to a nick translation reaction [133] (see Figure 2 and Figure 3) in which the displaced ssDNA tails are immediately cut and never exposed to the cellular environment unless some factor necessary for their processing is absent. Hence, when all factors necessary to process OFs are present in the cells, there should not be any net DNA synthesis, but it may be possible that, with a low frequency, regions of the lagging strand behind the replication fork are re-synthesized. A key step of the short flap OF processing pathway, which involves Fen1-PCNA-Pol δ, and is thought to be responsible of processing of the large majority of OFs, is the idling of Pol δ [47]. More recent and detailed in vitro millisecond studies using the Fen1-PCNA-Pol δ complex, confirmed that the 5’ end of the previous OF at a time acts as a barrier for PCNA-Pol δ mediated strand displacement so that it remains limited to the displacement of one nucleotide, which is immediately cut by Fen1 [49]. Moreover, in the same study, it was shown that the PCNA-Pol δ-Fen1 complex moves on processively, displacing one nucleotide from the 5’ end of the last OF, which is immediately cut by FEN1, so that the complex executes several iterative rounds of nick translation until the RNA primer is completely removed and the remaining nick is sealed [49] (Figure 2). 

Conceptually, this mechanism of OF processing through nick translation, excludes the formation of DNA flaps longer than one or two nucleotides, as well as direct RNA-DNA primer removal through RNase H2. If all the displaced nucleotides (or ribonucleotides) are immediately cut by FEN1 (or Exo1), also Dna2 activity would be dispensable. This is clearly not the case in vivo. It is not known if there are cellular pathways (or structural determinants in Pol δ) that regulate the switch between its 3’–5’ proof reading activity and its strand displacement activity. The structure of the *S. cerevisiae* Pol δ catalytic subunit (residues 68 to 985) has been determined with a resolution of 2Å [134]. The DNA polymerase domain of DNA polymerase δ and its 3’–5’ exonuclease domain are at a reciprocal distance of 45Å and the axis of the DNA molecule that is being synthesized is perpendicular to a line that connects the two domains of the enzyme [134]. Based on this structure, it was proposed that if the correct nucleotide is paired to the template nucleotide in the active site of Pol δ (Watson/Crick pairing), several direct or water mediated hydrogen bonds are created with the residues in the polymerization site. However, if a mismatch is present, the loss of these hydrogen bonds is believed to switch the binding of the template-primer DNA, from the polymerase to the exonuclease domain resulting in the switch from polymerization to proof-reading activity [134]. Although there are structural reasons to rationalize the switch from the DNA polymerization to the proof-reading activity of Pol δ, the in vivo determinants of the switch from normal DNA synthesis to DNA synthesis through strand displacement (when the 5’ of the last OF is encountered) are not known.

## 4. Extended Pol δ Strand Displacement and the Long Flap Pathway of Okazaki Fragment Processing in *S. cerevisiae*

The first evidence of a possible involvement of DNA helicase-mediated displacement events during OF processing was obtained in the fission yeast where it was shown that mutations in the *pfh1* gene suppress the thermo-sensitivity associated to the lack of *cdc24* functions [59]. Since Cdc24 was known to interact with Dna2, it was proposed that Pfh1 is involved in OF processing. Subsequently more detailed studies in *S. cerevisiae* have shown that the cell lethality induced by the absence of Dna2 can be suppressed by inactivating Pif1 or Pol 32 [53]. Based on this strong genetic interaction between Pif1 and Dna2, it was proposed that Pif1 and Pol 32 stimulate Pol δ-mediated strand displacement during lagging strand DNA synthesis leading to the creation of long DNA flaps that must be immediately processed by Dna2 to prevent DDR hyper-activation, cell cycle arrest and cell death [53,57]. Pol 32 is a DNA polymerase δ subunit required for Pol δ processivity during normal DNA synthesis [135]. DNA polymerase δ is composed of three subunits that contain PCNA Interacting Protein motifs (PIP), namely, Pol3 (catalytic subunit), Pol31 and Pol32 accessory subunits (see Table 1) [136]. Pol δ processivity during DNA synthesis depends upon the interaction with PCNA, which is achieved through the synergistic action of the three PIP motifs on its subunits [136]. Ablation of the PIP motif of Pol 32 (the only non-essential subunit of Pol δ), reduces the processivity of the PCNA-Pol δ complex in primer extension reactions in vitro [136]. This is in agreement with an early study in which it was shown that Pol δ is less processive without Pol 32 [135]. In this context, it is usually considered that *POL32* deletion reduces Pol δ processivity and its strand displacement potential. Recently, the high-resolution structure of PCNA in complex with a peptide of Pif1 carrying its PIP motif, has been resolved [98]. This data supports the idea that, through its interaction with PCNA, Pif1 can act as strand displacement factor for Pol δ [53,98]. In line with this hypothesis, it has been recently shown that the Pif1-PCNA interacting domain is specifically necessary for the in vivo lagging strand replication of DNA sequences that in silico analysis predict to fold into G-quadruplexes DNA structures [137]. Moreover, the PCNA interacting domain of Pif1 has been shown to be necessary to stimulate Pif1- and Pol δ-dependent strand displacement activity in vitro [94]. These results support the idea that Pif1 actions during DNA replication rely on its interaction with PCNA and that Pif1 could act as a strand displacement factor of Pol δ. However, it is not known whether point mutations in the Pif1-PCNA interaction domain of Pif1 can rescue the cell lethality associated with Dna2 depletion. From the initial reports of a strong genetic link between Dna2 and Pif1 [53,59], several in vitro studies started to dissect the roles of Dna2 and Pif1 in OF maturation. Both Dna2 and Fen1 can translocate on the ssDNA tail of a DNA flap in 5’–3’ direction and Fen1 can remove a Dna2 nuclease mutant that had translocated from the 5’ end tail to the junction between ssDNA and dsDNA at the base of a DNA flap, suggesting that Dna2 and Fen1 can act sequentially to process OFs [74,138]. Other in vitro reactions simulating OF processing and containing RPA, PCNA, Pol δ, Fen1 and Pif1 showed that Pif1 stimulates the displacement of a fraction of long DNA flaps (30 nts) without being influenced by the presence of a RNA-DNA initiator at their 5’ ends, leading to the formation of ssDNA-RPA nucleoprotein complexes that inhibit Fen1-mediated processing of the DNA flaps [50]. A subsequent study showed that Dna2-mediated removal of RPA from the long DNA flaps facilitates primer cleavage and processing by Fen1 [139], supporting a sequential mode of action of Dna2 and Fen1 in OF processing (Figure 2). This in vitro data is in good agreement with a later study in which it was shown that DDR activation causes lethality in Dna2 defective cells [57]. Importantly, since *rad27* cells are alive, it is reasonable to think that they accumulate flaps (or other aberrant DNA replication intermediates) that do not hyper-activate DDR to a level incompatible with cellular proliferation. In other in vitro reactions with Pif1, PCNA and Pol δ, strand displacement was extended to 2.9 kb and more then 10 kb, respectively, depending on the substrate used [94,98], suggesting that these three factors could form a molecular machine with high strand displacement activity similar to bacteriophage protein modules composed by a DNA polymerase and a DNA helicase. While it is known that Pif1 and DNA polymerase δ interact with PCNA [98,136], it is not known if in vivo Pif1-mediated strand displacement during OF processing requires its interaction with PCNA. These genetic and in vitro results suggest that long DNA flaps created by the Pif1-PCNA-Pol δ complex require the concerted actions of Dna2 and Fen1 in order to be processed and to avoid RPA binding to their 5’ ssDNA tails. In a more detailed in vitro study, using a reconstituted system that recapitulates OF processing, it was shown that, in most cases, the flaps (including the RNA-DNA initiator) are removed by Fen1, but a consistent fraction of OFs is displaced by Pif1, which is proposed to accelerate the rate of Pol δ-dependent strand displacement, leading to RPA binding to the displaced ssDNA tails and subsequent inhibition of Fen1 cutting [51]. In this case, the action of Dna2 was shown to be essential for the shortening of the DNA flap, removal of RPA and creation of a substrate (short flap), which can be cut by Fen1 to create a ligatable nick for ligation [51]. 

The precise extent of Pol δ-mediated in vivo strand displacement during lagging strand DNA synthesis is not known. There might be several advantages in having extended strand displacement events on the lagging strand. Firstly, DNA polymerase α does not have proof reading activity [140], and thus the DNA initiators are synthesized with low fidelity. If the Pol α-dependent DNA initiator on the lagging strand is displaced and synthesized by the more faithful Pol δ (during its extended and backward directed strand displacement DNA synthesis), lagging strand DNA replication will have a reduced load of mutations. Secondly, theoretically bulky obstacles that do not involve covalent links to the template strand (for example non-nucleosomal DNA-protein complexes and RNA-DNA hybrids in transcribed units) can be overcome and left behind the fork through continuous Pol α-mediated priming events and later removed, post-replicatively, by the backward directed strand displacement activity of the Pif1-PCNA-Pol δ complex [98] (Figure 4A). In this case, backward directed DNA synthesis refers to the direction of the DNA polymerase δ-mediated DNA synthesis on the lagging strand, which has a opposite direction compared to the direction of the moving replication fork (see Figure 1 and Figure 4). According to this model, bulky lagging strand obstacles do not induce fork pausing in Xenopus laevis [141] and Pif1 was shown to remove Rap1 at telomers, promoting the strand displacement activity of Pol δ, and to remove DNA-RNA duplexes in the R-loops, preventing genome instability at the tRNA genes [94,97]. R-loops are defined as ssDNA bubbles in which one of the two ssDNA filaments constituting the ssDNA bubble (or part of it) is annealed to an RNA molecule creating a DNA-RNA hetero-duplex [142]. To date, there are many indirect evidences that support the idea that R-loop structures are created during DNA transcription [142]. Importantly, R-loops are thought to be DNA replication obstacles that induce DNA replication fork pausing. If DNA replication across R-loops is not properly executed or regulated by several cellular factors, R-loops can become a source of genomic instability [97,143,144,145,146,147,148]. Hence, the potent and extended PCNA-Pif1-Pol δ mediated strand displacement activity on the lagging strand could be a mechanism to overcome DNA replication obstacles and avoid fork pausing (Figure 4A). Pol δ-dependent DNA synthesis through strand displacement could occur also during conservative break-induced replication (BIR), where Pif1 was proposed to promote D-loop migration [149,150,151,152] (Figure 4B). BIR is a specific type of DNA synthesis induced by a DSB [153]. In particular, when one of the resected DNA ends of a DNA double strand break is blocked and cannot be engaged in the homology search in the first step of DSB repair through homologous recombination, the other DNA end can promote strand invasion into the homologous sequence of the sister chromatid or homologous chromosome creating a D-loop structure where the 3’ end of the invading filament in the D-loop starts to be elongated through Pol δ-dependent DNA synthesis assisted by Pif1 (Figure 4B) [149,153]. In this case, Pif1 could have a double role: to promote D-loop migration and extension of the 3’ end of the DNA into the D-loop, and to promote strand displacement DNA synthesis that catalyzes the re-replication of the displaced strand during conservative BIR leading to a fully replicated duplex (Figure 4B) [149,150]. Thirdly, Pif1-PCNA-Pol δ-mediated strand displacement can readily remove DNA flaps that contain DNA hairpins that would otherwise inhibit Dna2-and Fen1-mediated processing [52] (Figure 4C). We would like to point out that problematic DNA flaps carrying DNA hairpins can be frequently created during lagging strand replication of telomeres, centromeres and other repetitive sequences in the genome and that these “folded-back flaps” could inhibit a proper processing and ligation of OFs. On the contrary, if folded-back 5’ ends of OFs are removed by Pif1-PCNA-Pol δ-mediated strand displacement, the subsequent cut of these displaced flaps by Dna2 could increase the efficiency and fidelity of OF processing in these repetitive regions (Figure 4C). Fen1, Dna2, Pif1, Pol δ and DNA ligase I interact, directly, with PCNA [64,78,98,154,155]. Extended Pol δ-dependent strand displacement events could imply additional backward directed lagging strand DNA synthesis events uncoupled from leading and lagging strand DNA synthesis at the replication fork branching point (see Figure 1, Figure 2 and Figure 4). These additional lagging strand DNA synthesis events behind the replication fork, with a direction opposite of the movement of the replisome, would displace the nucleosomes, which have been assembled on the newly replicated lagging strand DNA filament. There is good evidence both in vivo [156] and in vitro [28] that nucleosomes limit strand displacement synthesis by Pol δ. However, Pif1 and Rrm3 can promote DNA replication fork progression through nucleosomal and non-nucleosomal DNA-protein complexes [94,157,158,159]. It is possible that the complex PCNA-Pol δ-Pif1-Dna2 (or PCNA-Pol δ-Rrm3-Dna2) utilizes the energy from the ATP hydrolysis generated by the ATPase domains of Pol δ, Pif1 and Rrm3 to generate the driving force to displace the assembled nucleosomes from the lagging strand filament. It is also reasonable to hypothesize that the flexibility of OF maturation [131] and the dynamic usage of the short versus the long flap OF processing pathway is influenced by the structure of a hypothetical multi-protein complex composed by Pif1, Fen1, Dna2, PCNA and Pol δ and their relative three-dimensional positioning and interactions with the DNA template on the lagging strand. A graphical map of the direct physical interactions between the major factors involved in OF maturation is shown in Figure 5A. High resolution structures of *S. cerevisiae* Pol δ catalytic subunit [134], human PCNA in complex with FEN1 [160], human Pif1 helicase domain (and bacteroid full length Pif1) [161], *S. cerevisiae* Pif1 (residues 237–780) bound to polydT [162], and Dna2 [163] have been reported. However, in spite of this progress, it is not clear if macromolecular assemblies of these factors exist in vivo.

It is possible that, in the near future, new technical achievements in cryo-electron microscopy will help to elucidate the structure of these large “macromolecular machines”. Such structural information may also provide explanations for the underlying molecular mechanisms of previous biochemical or genetic evidence. For example, an in vitro study showed that a potent DNA helicase activity of Dna2 can be unleashed only if the nuclease domain is mutated [164]. In the light of the recent high resolution structure of Dna2, it is possible to explain this effect: based on the Dna2 structure, the 5’ end of a DNA flap should first enter the nuclease domain of the protein (so that it is degraded by the nuclease site); it very unfrequently reaches the helicase domain to trigger the ssDNA translocase or DNA helicase activities of Dna2 [163,164], unless the nuclease domain is inactivated.

In this context, in Figure 5B–E, different possible outcomes of the OF maturation process are schematized according to different hypothetical structural architectures of a macromolecular complex composed by Pif1, Fen1, Dna2, Exo1, PCNA and Pol δ. For example, the relative spatial positions (and interactions) of these factors (and their three-dimensional positioning towards the 5’ end of the last Okazaki fragment) would dictate if there will be an extended or reduced strand displacement and if the 5’ end of the last flap will be processed by FEN1, Exo1 or displaced by Pif1 and cut by Dna2. In this context, the structure of this hypothetical macromolecular assembly will also dictate if the 5’ end of the previous OF will enter the nuclease domain of Dna2, Fen1, Exo1, or if it will become substrate for Pif1, which will start to translocate on it in a reaction that may be stimulated by the advancement of Pol δ [74,99]. It is also possible that Pif1 translocation on the 5’ of the previous OF would exclude the ssDNA tail of the displaced flap from Pol δ-mediated 5’–3’ exonuclease activity. Moreover, the structure and/or composition of such macromolecular assemblies may also vary depending on the genome site in which DNA replication occurs, the stage of the S-phase or if the strand displacement activity of this complex is utilized in other cellular processes such as break induced replication (BIR) [98,150,152] (Figure 4B), error-free DNA damage tolerance [165,166], or other homologous recombination reactions. Additional investigations will also be required to clarify if there are cellular regulations of the extension of the DNA synthesis through strand displacement during lagging strand DNA synthesis and if there are changes of the structure of PCNA-Pif1-Pol δ complex during OF processing. Although OF maturation has been extensively studied in the last 25 years and the major framework of this cellular pathway is elucidated, the complex and intricate mechanisms and regulations of this process still require further investigations in the future.

## Figures and Tables

**Figure 1 genes-10-00167-f001:**
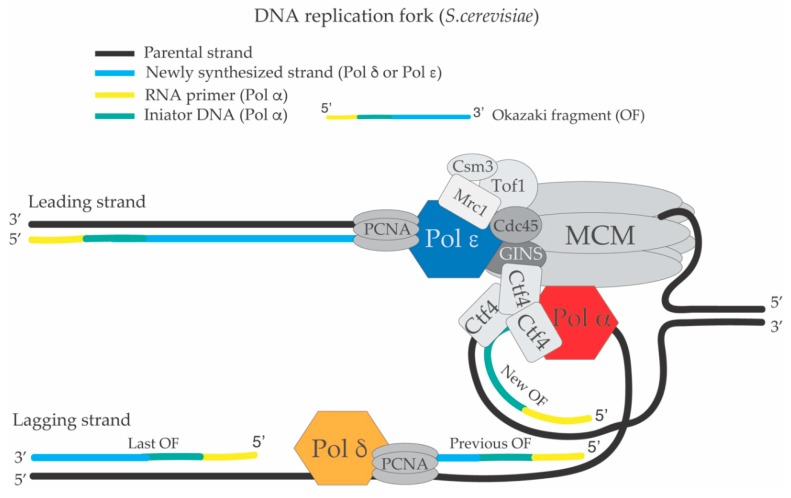
Schematic structure of the replisome and the DNA replication fork of *S. cerevisiae.* DNA polymerase ε synthesizes the leading strand continuously while DNA polymerase α-Primase and DNA polymerase δ carry out lagging strand synthesis in a discontinuous way creating Okazaki fragments (OFs). DNA polymerase α-Primase makes the RNA primer and synthesizes the DNA initiator fragment. After this event, Pol α falls off from the 3’ end of the growing OF at the replication fork and PCNA (Proliferating Cell Nuclear Antigen) is loaded by the RFC (Replication Factor C) complex leading to a DNA polymerase switch between Pol α and Pol δ on the lagging strand. The growing 3’ end of the previous OF, which is being synthesized by Pol δ will encounter the 5’ end of the last OF, which contains the RNA-DNA initiators segments previously synthesized by Pol α. This encounter will lead to the processing and maturation of the OF (see Figure 2).

**Figure 2 genes-10-00167-f002:**
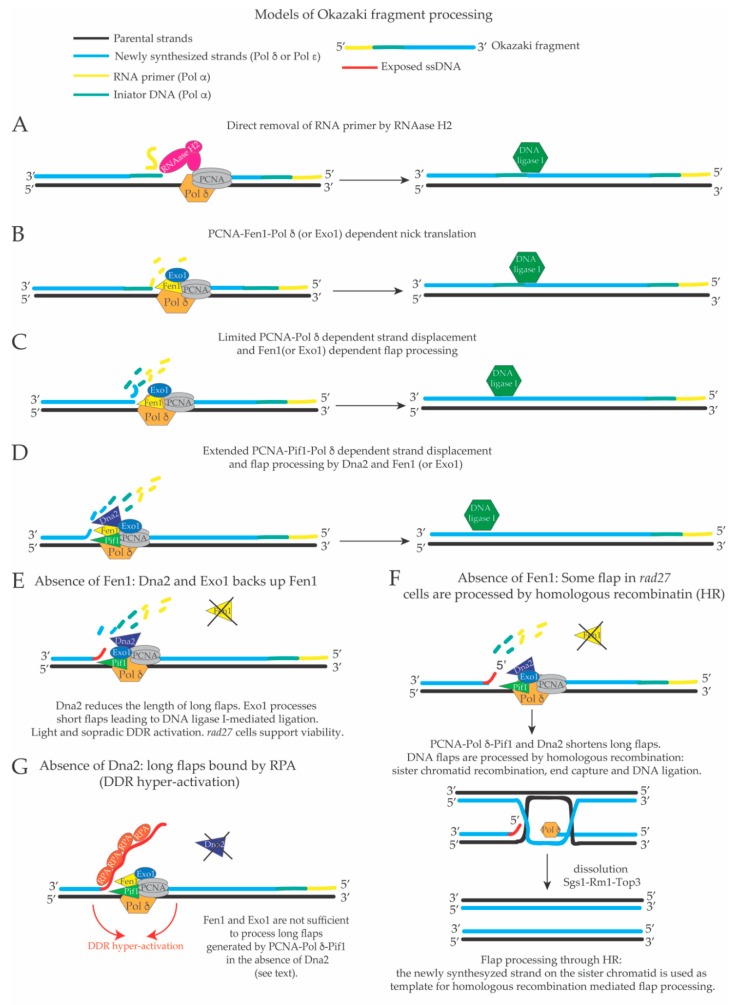
Possible mechanisms of Okazaki fragments (OF) maturation in *S. cerevisiae*. (**A**) Direct removal of the RNA primer by RNase H2 combined with limited Pol δ-mediated DNA synthesis leads to the formation of a ligatable nick, which is a substrate for DNA ligase I. (**B**) Limited strand displacement of the 5’ end of the last OF by Pol δ (due to DNA polymerase idling) displaces a single nucleotide that is cut by Fen1 or Exo1. Iterative cycles of Pol δ-Fen1 or Pol δ-Exo1-dependent nick translation will remove the RNA primer and create a ligatable nick for DNA ligase I. (**C**) More cycles of nick translation will remove also the initiator DNA previously synthesized by Pol α. (**D**) Extended Pif1-PCNA-Pol δ-dependent strand displacement DNA synthesis proceeds beyond the initiator DNA. The displaced 5’ end cannot be cut by Fen1 and Exo1 and requires processing by Dna2. Once Dna2 has reduced the length of the displaced tail, short flaps become substrates for Fen1 or Exo1, which will create a ligatable nick for DNA ligase I (note sequential actions of Dna2 and Fen1). (**E**) In the absence of Fen1, Pif1-PCNA-Pol δ-dependent long flaps are shortened by Dna2 to a length that only sporadically allows the binding of RPA and DDR activation (also because Exo1 can back up Fen1 on short flaps, so that *rad27* cells can support viability; see the text). (**F**) Small flaps produced in the absence of Fen1 may also be repaired through homologous recombination with the sister chromatid. This mechanism could explain why *RAD27* mutations are lethal in combination with homologous recombination mutations. (**G**) In the absence of Dna2, Pif1-PCNA-Pol δ -dependent long flaps are not processed, and ssDNA bound by RPA remains exposed, Fen1-mediated cutting is inhibited and DDR hyper-activation is triggered. Exo1 or RNase H2 should not be able to cut long DNA flaps in *dna2* cells and does not prevent DDR hyper-activation in *dna2* defective cells. DDR activation in Dna2 depleted cells leads to cell cycle arrest and cell death, strongly suggesting that the products of OF maturation in the absence of Dna2 are not compatible with viability.

**Figure 3 genes-10-00167-f003:**
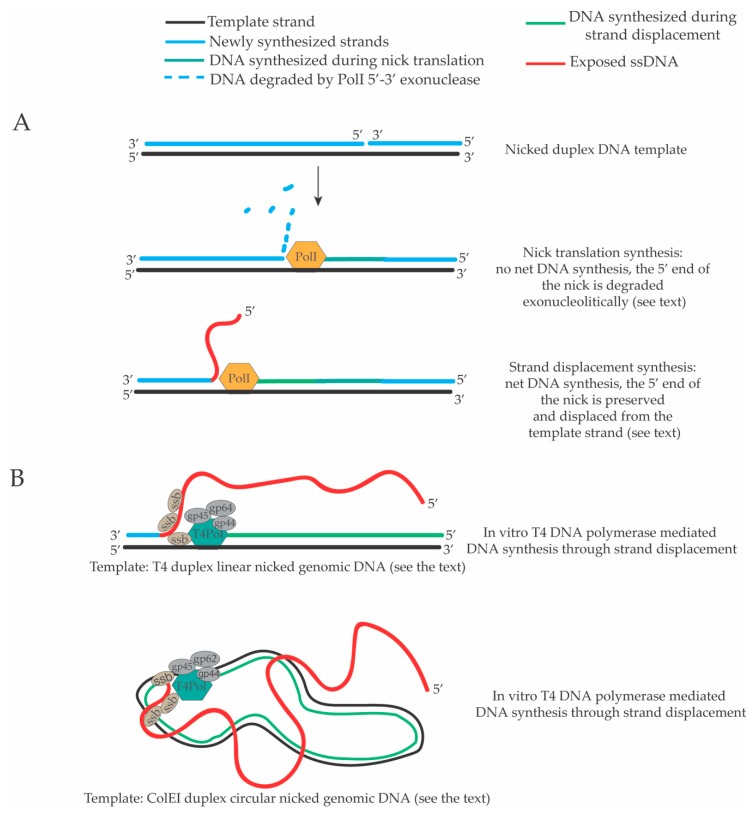
DNA synthesis through strand displacement in bacterial and bacteriophage DNA polymerases. (**A**) *E. coli* DNA polymerase I (PolI) can carry out DNA synthesis through strand displacement on a nicked duplex linear DNA molecule. When PolI starts to extend the 3’ end of the nick there is a first phase of nick translation where the 5’ end of the nick is degraded by its 5’–3’ exonuclease activity. In this phase, there is no net DNA synthesis. A second phase of DNA synthesis through strand displacement follows in which the 5’–3’ exonuclease activity is inhibited and the 5’ end of the nick starts to be displaced in a ssDNA tail [111,112]. (**B**) The T4 DNA polymerase (T4Pol, gp43 protein) together with the T4 single strand binding protein (ssb, gp32 protein) and other structural factors (gp62, gp45 and gp44) can carry out extended DNA synthesis through strand displacement (in the absence of the T4 DNA helicase, gp41 protein) either on linear duplex or circular duplex nicked DNA templates. The activity of the T4 Pol on these substrates produces long displaced ssDNA tails [114].

**Figure 4 genes-10-00167-f004:**
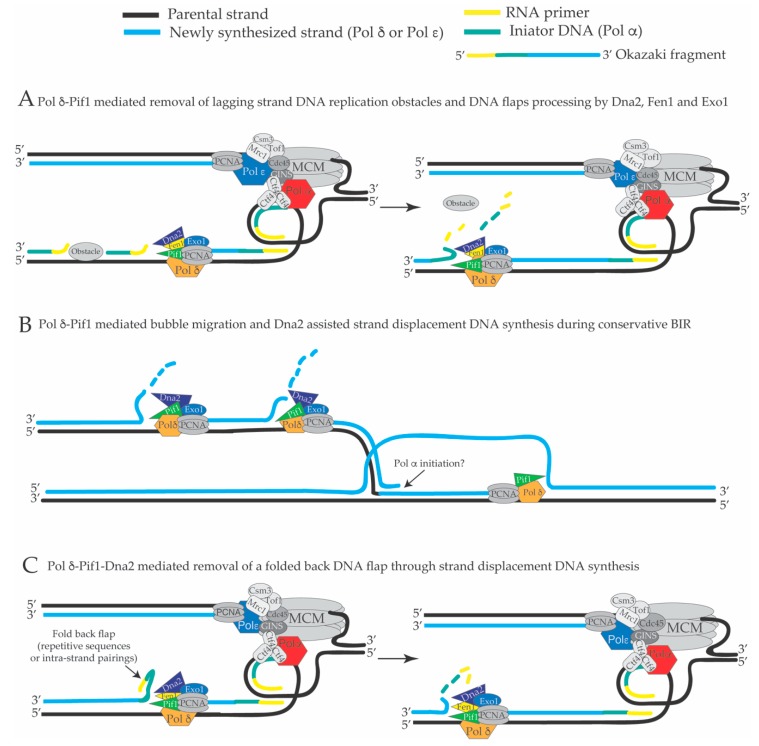
(**A**) Backward directed Pif1-PCNA-Dna2-Pol δ mediated strand displacement DNA synthesis and flap cutting may be utilized to remove DNA replication obstacles such as non-nucleosomal DNA-protein complexes and R-loops created by the transcription units to counteract DNA replication fork pausing (see the text). (**B**) Pif1-Pol δ mediated D-loop bubble migration and extended strand displacement DNA synthesis coupled to Dna2-dependent cutting of the resulting DNA flaps during conservative BIR (break-induced replication) contribute to create a fully replicated duplex. (**C**) Backward directed Pif1-PCNA-Dna2-Pol δ-mediated strand displacement DNA synthesis and flap cutting may be utilized to remove flaps carrying DNA hairpins during lagging strand DNA replication of repetitive sequences.

**Figure 5 genes-10-00167-f005:**
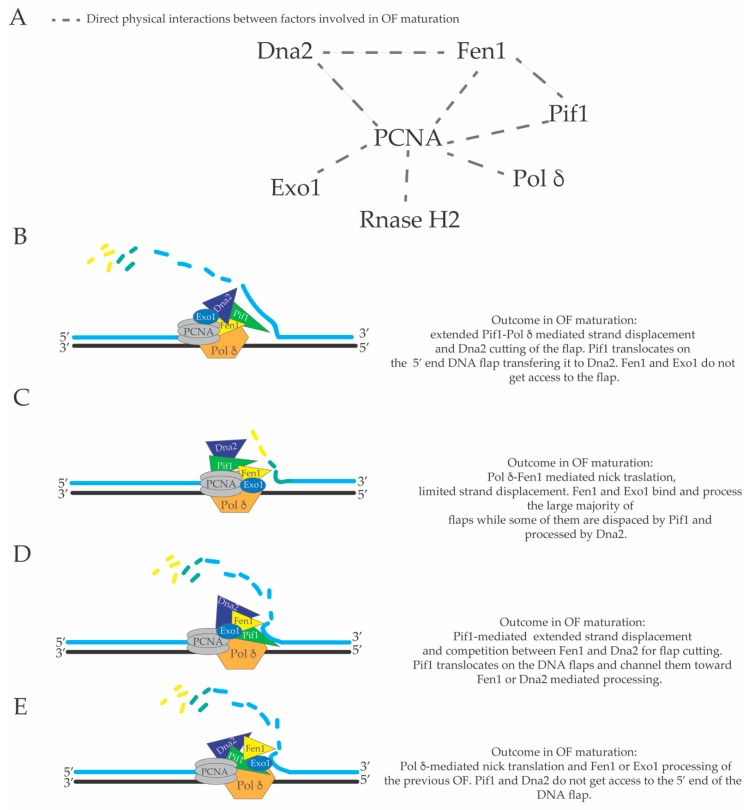
(**A**) graphical map of the direct physical interactions of factors involved in OF processing. Different mechanisms of OF processing dictated by different structural architectures of the complex Pif1-PCNA-Dna2-FEN1-Pol δ macromolecular assembly (**B**–**E**).

**Table 1 genes-10-00167-t001:** Protein factors, complexes and genes involved in OF maturation.

*Saccharomyce Cerevisiae*		*Schizosaccharomyces Pombe*		Human	
Protein(s)	Gene(s)	Protein(s)	Gene(s)	Protein(s)	Gene(s)
PCNA	*POL 30*	PCNA	*pcn1* ^+^	PCNA	*PCNA*
DNA Polymerase δ (Pol 3-Pol 31-Pol 32)	*POL 3*, *POL 31* and *POL 32*	DNA Polymerase δ (cdc6-cdc1-cdc27)	*cdc6*^+^, *cdc1*^+^ and *cdc27*^+^	DNA Polymerase δ (p125-p50-p68-p12)	*POLD1, POLD2,* *POLD3 and POLD4*
Pif1	*PIF1*	pfh1	*pfh1* ^+^	Pif1	*PIF1*
Dna2	*DNA2*	dna2	*dna2* ^+^	Dna2	*DNA2*
Exo1	*EXO1*	exo1	*exo1* ^+^	Exo1	*EXO1*
Rad27	*RAD27*	rad2	*rad2* ^+^	FEN1	*FEN1*
RNase H2 (Rnh201-Rnh202-Rnh203)	*RNH201, RNH202* and *RNH203*	RNase H2 (Rnh201-Rnh202-Rnh203)	*rnh201*^+^, *rnh202*^+^ and *rnh203*^+^	RNase H1	*RNASEH1*

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
