# Peer review of "DNA Replication Through Strand Displacement During Lagging Strand DNA Synthesis in *Saccharomyces cerevisiae"

_genes, 2019, doi:10.3390/genes10020167_

Round 1
Reviewer 1 Report
Dear Authors,
I read your review with great interest. You succeeded very well in compiling a very informative and good review on a rather complicated subject. Congratulations!
Here are my comments (in order of the lines)
L24: origin licensing in S.cerevisiae
L33: with CDK-phosphorylated Sld3..
L39: events rely on…
L40: delete the word “cooperate”
L51: creating Okazaki fragments (OFs)- It may help to label the OFs like “Previous OF” & “New OF” in Figure 1
L64: mechanisms of…. forks are…
L66: processivity of Pol alpha. (start new sentence) Pol alpha falls off…
L67: is thereby expelled
L69: Delete “Besides Ctf4” – start with “The pausing…”
L72: important when the DNA replication fork…
L73: slows down or stalls to collaborate with..
L74: and facilitate fork rotation
L75: of DNA synthesis (delete “the”)
L76: super-coiling that is DNA accumulation ahead of the fork and to catenate DNA behind…
L80: maturation during which…
L83: in the genome, deletions.. (delete “then”)
L88: ligase 1 all participate in the…
L106: become a substrate…
L108: it would be important at this stage in the review to highlight that the extent of DDR (use DNA Damage Response the first time the abbreviation is used) correlates with the amount of singles-strand DNA covered by RPA as this activates the DDR master kinase Mec1 (Rad3, ATR) (+ reference).
L110: gene that encodes DNA Ligase I
L114: ad is stimulated by PCNA
L115: and is necessary for OF..
L118: participates in the processing of OFs.
L120: replace “Excluding” with “Besides”
L121: participates in OFs processing…
Figure 2B (label text) nick translation
L124: Okazaki fragment (OF)
L125: by RNAseH2
L126: be a substrate of…
L132: Fen1 or Exo1
L133: substrate for Fen1 (delete “also”)
L145: delete “in the absence of DNA2.”
L146: full stop after complex – start new sentence The catalytic subunit…
L147: is related to…
L151: slowly growing, the double mutant (delete”in S.cerevisiae”)
L152: Rad27 is dead. (better than lethal as this term has already been frequently used) After this sentence an explanation of these genetic interactions is missing.
L152: Rad27/Fen1 deletion shows…
L154: it would be clearer to say that the dependency on HR … supporting the idea that HR backs up loss of RNAseH2.
L154: please include some information on the remaining two subunits of RNAseH2
L155: lethality when combined..
L157: replace “Altogether” with “Taken together”
L159: delete the sentence “However, none of these three…”
L161: please introduce the mutations dna2-1 and dna2-2
L170 (throughout the manuscript) carried out not carried on
L170: network of overlapping pathways. (delete the rest of the sentence)
L175: the sentence referring to S.pombe Pfh1 could be deleted as it doesn´t add additional information. It may be worth including a table in the review listing the discussed S.c. genes & proteins together with the S.pombe and human para/homologous.
L177: associated with loss of DNA2 or Cdc42. (Cdc42 needs to be introduced here)
L179: creates
L179: you could add here a sentence like ”A very similar interaction exists in the fission yeast S.pombe.”
L180: is due to the activation of the DNA damage response DDR by RPA coated, long single-stranded flaps.
181 mutants trigger
L183: intermediates confirmed in germinated… S.pombe spores… in S.cerevisiae. (delete “cells”)
L185: is similar in the two reports
L186: this sentence is unclear
L189: delete “leading…..at the”
L198: whereas less frequent
L202: cause increased strand displacements that are lethal..
L23: suggesting an interplay
L204: role of its.. (delete “of Pol delta”)
L206: which prevents the creation.. favours thereby Fen1 or Exo1..
L210: While Pol delta.. (delete “Of interest”)
L212: inhibited by long ssDNA tails..
L215: When Pif1 is added..
L215: a conclusion is missing why pol delta becomes insensitive to growing ssDNA tails.
L216: a conclusion is missing after “to 30 – 110 nts.”
L225: delete the sentence “In this section…”
L230: polymerization when they…
L258: ColE1 plasmid DNA..
L267: synthesis revealed the presence..
L271: isomerization do occur..
L272: frequent products of..
L300: These results led to the concept that…. Composed of a potent DNA polymerase.. (delete text in between)
L308: (: is able to carry out..
L309: to produce.
L309: Like DNA polymerase delta, other eukaryotic … can also carry out extended..
L312: synhesises large..
L315: eukaryotes is stimulated
L316: please introduce PARP1 here
L326: and long flap..
L334: delete “mediated strand displacement” and “From this point of view” (start sentence with OF maturation (in wild type conditions) can therefore..
L337: necessary for
L337: Hence, when all factors… are present in the cell, there..
L339: synthesis. It may however be possible hat, with a low frequency..
L346: the previous OF at a time, which..
L350: excludes the formation….nucleotides as well as direct RNA-
L363: domain resulting in the switch
L373: delete “and other mitochondrial DNA repair processes.”
L374: Subsequently, more..
L389: as a strand displacement (delete “sort of”)
L396: known whether point mutations
L401 tail to the junction..
L410: fen1/rad27
L413: 2.9 kb
L414: from a molecular machine
L428: firstly (secondly) may be better than first (second)
L435 events and later removed
L436: please explain the backward directed strand displacement activity here
L438: please introduce R-loops
L444: please explain BIR here
L445: folded back (you may consider the term “hairpin” here)
L449: OFs are removed by..
L463: clear whether macromolecular
Figure 4A: DNA2, Fen1..
Figure 4C: Folded back flap
In figure legend 4 you use 4A, 4B… A, B… is sufficient to label the subpanels
L470: full name for BIR
L474: replace “big” with “large”
L476: evidence
L478: delete “recent”
Figure 5C: translation and replace “few” with “some”
L504: replace “regulations of” with “changes to the structure”
Author Response
Specific comments to Reviewer 1.
We inserted all the suggested corrections and citations in the manuscript and modified
several sentences as suggested. We like to thank you for your help in improving the
manuscript and for your enthusiasm on the value of this review.
We included in the text all the minor changes suggested and addressed the major points as
described below.
Major points:
1) We now highlight that the extent of DDR activation correlates with the amount of singlestrand
DNA covered by RPA as this activates the DDR master kinase Mec1 (Rad3, ATR)”.
We also added relevant references.
2) We now added an extended paragraph to provide possible explanations of why Rad27,
Exo1 and RNase H2 defective cells are alive and why the double mutants rad27 exo1 and
rad27 rnh35 cells are dead even though HR and Dna2 are envisaged to be fully functional in
these cells. We propose that two redundant pathways (Rad27 and Exo1-RNase H2) exist
and must modify the DNA flaps during OF maturation so that the flaps can become
substrates for Dna2 or HR.
3) We now mention in the text (as suggested by the reviewer), that mutations in RNH35 or
other genes encoding for RNase H2 subunits induce cell lethality when combined with HR
mutations.
4) The sentence referring to S. pombe Pfh1 was removed and instead, as the reviewer
suggested, we added a table (Table 1) listing the factors contributing to OF processing and
their nomenclature in different model systems discussed.
5) We included additional information about the RNase H2 subunits.
6) We described which aminoacidic substitutions are present in the dna2-1 and dna2-2 alleles
and how they impact on Dna2 activities.
7) Cdc24 function and relationships with Dna2 were added to the text.
8) We discussed how Pol d (when bound to PCNA) could become insensitive to the presence
of ssDNA tails.
9) We now describe in the text why we refer to the Pol d-mediated lagging strand synthesis
as “backward”-directed DNA synthesis.
10) We now describe in the text the structure of R-loops, how they are created and give some
general information about their relationships with DNA replication and genome instability.
11) We introduced better PARP1 in the text mentioning its biochemical activity and known
cellular roles including the very important genetic interaction with BRCA mutations.
12) We now introduce in the text more precisely what BIR is, when it is activated, the role
of Pif1 in BIR and the hypothetical new function of the complex PCNA-Pif1-Dna2-Pol d in
the conservative DNA synthesis through BIR.
13) As requested and mentioned above, we added a new table (Table 1), which includes all
the protein factors and protein complexes involved in OF maturation in S. cerevisiae, S. pombe
and mammalian cells.

Reviewer 2 Report
This review by Giannattasio and Branzei describes lagging strand processing and maturation in S. cerevisiae. The review offers detailed descriptions of the various pathways that are employed to remove the RNA/DNA primers synthesized by Pol alpha prior to ligation, with an emphasis on strand displacement synthesis catalyzed by Pol delta. As it stands I feel that the review provides limited new insights or ideas into the mechanisms of lagging strand processing and is quite challenging to read in places. There is also a large section on the strand displacement activity of prokaryotic DNA polymerases, which although interesting, seems rather unnecessary given the focus on S. cerevisiae. There are also several inaccuracies, especially in the section describing replisome assembly, including the citation of a retracted manuscript. The authors should consider the following comments:
1. The description of replisome assembly (lines 21-76) needs to be significantly reworked. Some of the descriptions are misleading and inaccurate, the referencing is incomplete, and in one instance a retracted paper is cited.
a. The role of DDK is not accurately reported. The authors write “Once the CMG complex and DNA polymerase eare loaded onto an ARS, not fully understood events relying on the single strand binding protein RPA and DDK-dependent phosphorylation of MCM, cooperate to unwind the DNA around the replication origin8.” This is completely inaccurate and the paper that is cited has actually been retracted. Rather, DDK phosphorylation of Mcm4/6 generates binding sites for Sld3 (Deegan et al, 2016, EMBO), which recruits Cdc45. This work should be cited. Currently, there is very little evidence to suggest that DDK phosphorylation has additional functions in CMG assembly and activation. MCM is the only essential DDK target for replisome assembly (Yeeles et al, 2015, Nature) and cryo-EM structures of DDK-phosphorylated MCM show that there are almost no structural changes (outside the Mcm4/6 N-terminal tails) compared to the unphosphorylated complex (Abid Ali et al, 2017 Nat comms).
b. Lines 43-45 the authors write “upon phosphorylation, MCM changes its conformation and encircles……..” This is misleading. As stated above, phosphorylation per se results in little change to the MCM double hexamer. Rather phosphorylation of MCM together with S-CDK phosphorylation of Sld2 and Sld3 triggers the recruitment of firing factors which remodel the MCM double hexamer to form CMG. How this remodelling occurs is not fully understood.
c. Currently there is no discussion about the role of Mcm10 in helicase activation despite the protein being essential for this process (Yeeles et al, 2015, Nature; Douglas et al, 2018, Nature). The role of Mcm10 should be included in a revised manuscript.
d. The relationship between Ctf4 and Pol alpha needs to be more clearly defined. Currently it is stated that Pol alpha is recruited by Ctf4 (line 35) but also directly by MCM (line 45). Evidence from both in vitro (Yeeles et al, 2015, Nature, Georgescu et al, 2015, Elife) and in vivo (Evrin et al, 2018, EMBO) systems has demonstrated that Ctf4 is dispensable for priming. Rather, it seems Pol alpha binding to Ctf4 is important for parental histone transfer. How Pol alpha is recruited for priming is yet to be fully resolved.
e. One of the key functions of Mrc1-Csm3-Tof1 is to facilitate normal rates of fork progression. This should be mentioned when these proteins are introduced.
f. I feel that there is too much emphasis placed on the paper that suggests Pol delta may replicate both strands (lines 62 and 63), which will just be confusing non-replication experts. Pol delta synthesis of the leading strand certainly is not the prevailing view in the field and has no support from in vitro DNA replication reactions – multiple systems clearly show that leading-strand replication by Pol epsilon is required for normal rates of fork progression (Yeeles et al, 2017, Mol Cell; Georgescu et al, 2014, NSMB; Devbhnadari et al, 2017 Mol Cell; Aria and Yeeles 2018, Mol Cell).
2. There is good evidence both in vivo (Smith and Whitehouse, 2012, Nature) and in vitro Devbhandari et al 2017 Mol Cell) that nucleosomes limit strand displacement synthesis by Pol delta. This is not currently discussed but should be included in a revised manuscript. Similarly, there is evidence for the involvement of Pif1/Rrm3 in strand displacement though nucleosomes in vivo that should be cited (Osmundson, 2016, NSMB)
3. Lines 78-79 – the RNA part of the primer, RNA initiator, is abbreviated to RNAi. This should be changed given the widespread use of RNAi as an abbreviation of RNA interference.
4. Lines 78-79 – the RNA initiator is stated to be about 20 nucleotides. The work that is cited is based on experiments in E. coli. Given that the review is specifically about lagging-strand processing in S. cerevisiae these citations don’t seem appropriate. Line 46 says primers of 40 nt are made. This seems to inaccurate as well. Appropriate S. cerevisiae references should be included. The length of the RNA is thought to be around 10 nt.
5. Generally, the use of Rad27/FEN1 may be confusing to non specialists. It would be much clearer is FEN1 is used to describe the protein throughout (perhaps apart from when specifically referring to the S. cerevisiae gene)
6. Line 163-164. What do the dna2-1 and dna2-2 mutations do to protein function?
7. Line 178 - When Cdc24 is introduced for the first time there is no explanation of what the protein does.
8. Line 340 – The use of the term re-replicated might be confusing given that it is frequently used to describe the firing of new origins on regions of the genome that have already been replicated.
9. Lines 434-440 – The pathway shown in figure 4A is described as ‘re-priming’. However, it is essentially just a continuation of normal lagging-strand priming, rather than a specific re-priming pathway. This pathway is suggested to be a mechanism to remove bulky obstacles and prevent forks pausing. However, evidence from Xenopus extracts suggests that bulky lagging-strand obstacles do not inhibit fork progression (Fu et al, 2011, Cell).
Author Response
Specific comments to the reviewer 2.
We like to thank the reviewer for very useful suggestions on how to improve the
quality of our manuscript.
We apologize for the citation of a manuscript that was recently withdrawn and for
not discussing the role of nucleosome displacement during replication and OF processing.
We also agree with the reviewer that the first part of the manuscript on origin activation,
replisome assembly and structure was excessively condensed and over-simplified. We have
thoroughly revised this part according to the reviewer’s suggestions. We also added a first
paragraph at the beginning of the text of the review in which we clarify that, due to space
limitations, the readers are invited to other recent reviews for the subject of replication
initiation regulation. We inserted in the text all the minor changes and references suggested
and we addressed the major points as described here below.
Major points
1) The description of replisome assembly has been significantly reworked in the spirit of the
reviewer’s suggestions. All the citations proposed by the reviewer have been included and
discussed in the text.
2) We now carefully describe in the text that DDK-mediated phosphorylation does not
induce a change in MCM conformation, but it recruits Sld3 and Cd45 to MCM and that
MCM is the only DDK substrate for origin firing.
3) We added a sentence on the fundamental role of MCM10 in CMG activation.
4) We mention in the text that Ctf4 does not seem to be essential for priming but it appears
to have a more defined role in transferring parental histones to the newly synthesized
lagging strand filament. We included the corresponding citations.
5) We added all the citations suggested by the reviewer that support a predominant role of
Pold for lagging strand synthesis and of Pole for leading strand synthesis. This is how we
intended to describe their roles in the first place.
6) We now state in the text that DNA synthesis through strand displacement mediated by
the PCNA-Pif1-Dna2-Pol d complex may act to displace nucleosomes on the lagging strand.
We added the citations suggested by the reviewer regarding the role of Pif1/Rrm3 in this
process.
7) As suggested by the reviewer, we removed the term re-replicated from the manuscript
and we stated that continuous priming by Pola may be used to overcome lagging strand
obstacles that do not necessary induce fork pausing. We added the corresponding citation
suggested by the reviewer that shows that in Xenopus laevis some specific lagging strand
obstacles do not induce fork pausing.
However, we feel that the section of the manuscript focused on prokaryotic protein modules
with potent strand displacement activity is important for our manuscript. It highlights the
concept that protein modules composed by a DNA polymerase and a DNA helicase can
carry out extended strand displacement DNA synthesis and provides a framework for our
proposal that PCNA-Pif1-Dna2-Pol d complex works in a similar way to these prokaryotic
and bacteriophage protein modules. We also feel that providing a historical perspective on
how DNA synthesis through strand displacement was discovered is important because it
allows us to introduce what this specific type of DNA synthesis is and how it works. Based
on these reasons, we chose to retain this section in our manuscript.
Round 2
Reviewer 2 Report
I thank the authors for addressing our comments on the original manuscript. I think the manuscript is now ready for publication.